# Simple Algorithms for Bad Triangle Transversals with Applications to Correlation Clustering

**Florian Adriaens** [1]  **Nikolaj Tatti** [1]

## Abstract

The Bad Triangle Transversal (BTT) problem asks for the smallest set of edges that need to be removed from a given signed graph, so that the resulting graph does not have a bad triangle. Here, a bad triangle is a triangle with exactly one negative edge. Several 2-approximations for BTT are proposed in this paper. On the hardness side, we show that BTT is NP-hard to approximate with factor better than $\frac{2137}{2136}$ on complete graphs. Our reduction also works for Correlation Clustering (CC), the Cluster Deletion problem (CD) and the Minimum Strong Triadic Closure problem (Min-STC). Lastly, we show that the BTT and CC optima are within a factor of 3/2 in complete graphs, by describing a pivot procedure that transforms transversals into clusters.

## 1. Introduction

A signed graph is a graph where every edge has a positive or negative label. Such graphs model a variety of different scientific phenomena. For example, they model the ground-state energy of Ising models in physics (Kasteleyn, 1963). Other natural applications are community detection in complex networks (Traag & Bruggeman, 2009), or in opinion dynamics, when the beliefs of individuals are influenced based on negative or positive interactions (Shi et al., 2016).

An important aspect in the edge formation in signed social networks is the sign of triangles, which has its roots in structural balance theory from social psychology (Cartwright & Harary, 1956). A prevalence of balanced triangles—those with an odd number of positive labels—has been observed in real-life social networks (Leskovec et al., 2010). Many methods for link and sign prediction try to capitalize on this, see for example the survey of Tang et al. (2016).

For *clustering* signed graphs, a widely popular approach is Correlation Clustering (CC) (Bansal et al., 2002). In CC the goal is to partition the vertices of a graph such that the number of positive inter-cluster edges and negative intra-cluster edges is minimized. CC has widespread applications ranging from biology to document clustering and computer vision (Bonchi et al., 2022).

Recently, connections between CC and bad triangle covering problems have been studied (Veldt, 2022; Bengali & Veldt, 2023; Makarychev & Chakrabarty, 2023; Cao et al., 2024b; Balmaseda et al., 2024; Fischer et al., 2025; Veldt, 2026). Here, a *bad* triangle is a triangle with exactly one negative edge. A *transversal* or *cover* is a set of edges that intersect all bad triangles.

In this paper, we provide several novel approximation algorithms and inapproximability results for the Bad Triangle Transversal problem (BTT). This problem asks to find a smallest-sized cover for a given signed graph. As we will see, the BTT problem is intricately related to the CC objective on complete signed graphs.

### 1.1. Results for general graphs

Our main contribution is two novel 2-approximations for BTT. Before providing any additional details, let us first discuss two existing baseline algorithms.

First, note that including all edges from any maximal set of edge-disjoint bad triangles immediately gives a 3-approximation. This holds because BTT is a special case of vertex cover in 3-uniform hypergraphs, which can be approximated by maximal matching. We will refer to this algorithm as the *standard 3-approximation*. All previous work aimed at solving BTT, or weighted variants thereof, utilize either the standard 3-approximation or a similar variant based on weighted vertex cover in 3-uniform hypergraphs (Grüttemeier & Morawietz, 2020; Veldt, 2022; Bengali & Veldt, 2023; Balmaseda et al., 2024; Oettershagen et al., 2025; Veldt, 2026; Oettershagen et al., 2026).

Second, although seemingly not mentioned in the literature, a direct observation is that the 2-approximation of Krivelevich (1995) from the 1990s for covering triangles in *unsigned* graphs also works for 2-approximating BTT.

[1]University of Helsinki, Finland. Correspondence to: Florian Adriaens <florian.adriaens@helsinki.fi>.

After solving the standard linear program (LP) relaxation of the covering problem, the algorithm of Krivelevich proceeds as follows: Iteratively remove edges with high LP-value, then re-solve the LP on the remaining graph. Repeat until a stopping criteria is satisfied. However, this algorithm is slow since it needs to solve $\mathcal{O}(m)$ cover LPs in the worst-case, where $m$ is the number of edges.

Instead, our 2-approximations for BTT directly round the optimal $\text{LP}_\triangle$ values into a feasible cover. Here, $\text{LP}_\triangle$ denotes the bad triangle cover LP which is defined in Section 3.1. Our algorithms are both simpler and faster than the method from (Krivelevich, 1995), since $\text{LP}_\triangle$ needs to be solved only once and our rounding procedures are quite simple. They are specifically designed for signed graphs, in the sense that they do not work for covering triangles in unsigned graphs.

To complete the picture in general graphs, Theorem 4.2 states that an approximation ratio of two is tight. Indeed, a straightforward reduction shows that BTT is as hard to approximate as vertex cover, meaning that a better than 2-approximation is UGC-hard (Khot & Regev, 2008).

## 1.2. Results for complete graphs

BTT is particularly interesting on complete graphs, since it becomes identical to the MinSTC+ labeling problem (Section 1.2.1) and it has a strong connection with the CC problem (Section 1.2.2).

On complete graphs Cao et al. (2024b) showed how to $(1 + \epsilon)$-approximate $\text{LP}_\triangle$ in time $\widetilde{\mathcal{O}}(\epsilon^{-7}m^{3/2})$, where $m$ is the number of positive edges. Since our randomized rounding method (Algorithm 3) also works on approximately optimal $\text{LP}_\triangle$ solutions, combining both methods leads to Theorem 1.1. Interestingly, its running time nearly matches the $\mathcal{O}(m^{3/2})$ time required for finding a maximal set of edge-disjoint bad triangles (Cao et al., 2024b), which is the time required for the standard 3-approximation.

**Theorem 1.1.** *For complete graphs with $n$ nodes and $m$ positive edges, there exists a randomized algorithm for BTT with a $(2 + \epsilon)$ approximation guarantee in expectation, in $\widetilde{\mathcal{O}}(\epsilon^{-7}m^{3/2})$ time. This algorithm can be derandomized in an additional $\widetilde{\mathcal{O}}(n^2)$ steps.*

Next, we present our main hardness result, which uses a gap-preserving reduction from a 2SAT minimization problem.

**Theorem 1.2.** *For complete graphs, BTT is NP-hard to approximate with factor better than $\frac{2137}{2136}$.*

An interesting observation is that this reduction also gives the same inapproximability result for three related problems: CC, the Minimum Strong Triadic Closure (MinSTC) problem (Sintos & Tsaparas, 2014), and the Cluster Deletion (CD) problem (Shamir et al., 2004).

Theorem 1.2 leaves open the possibility of a constant-factor approximation with factor smaller than 2 on complete graphs. Such method cannot be based on $\text{LP}_\triangle$ since its integrality gap is at least 2, even when restricted to complete graphs (Lemma 4.1). Yet, we believe it exists, seeing that similar improvements were recently discovered for CC (Cohen-Addad et al., 2022).

*Remark* 1.3. BTT on complete graphs has two other existing approximations, aside from the baselines discussed in Section 1.1. First, the well-known pivot algorithm from Ailon et al. (2008) is a 3-approximation. This follows from its proof (Ailon et al., 2008), which relates the mistakes made by pivot to the dual program of $\text{LP}_\triangle$. Secondly, Cao et al. (2024b) describe an algorithm that gives a 2.4 approximation. Although both methods were originally designed for approximating CC, they generate a feasible cover while charging the mistakes to $\text{LP}_\triangle$. Therefore, they also approximate BTT.

### 1.2.1. CONNECTIONS TO MINSTC+

Sintos & Tsaparas (2014) introduced the MinSTC+ problem as one of their edge labeling problems related to the strong triadic closure principle, with the goal of inferring strength of ties in social networks (Sintos & Tsaparas, 2014; Adriaens et al., 2020; Matakos & Gionis, 2022; Oettershagen et al., 2025). Prior work showed that MinSTC+ is identical to BTT on complete graphs (Grüttemeier & Morawietz, 2020; Veldt, 2022). Thus, our results directly apply to MinSTC+.

Algorithm 3 also 2-approximates *weighted* variants of MinSTC+. Several of these weighted variants have been studied in the literature recently, such as the LambdaSTC problem (Bengali & Veldt, 2023) and the MinSTC+ problem in temporal graphs (Oettershagen et al., 2025). These works use either the standard 3-approximation or a 3-approximation for weighted vertex cover in 3-uniform hypergraphs. Theorem 1.1 improves this to a near 2-approximation in comparable running time as the standard 3-approximation.

### 1.2.2. CONNECTIONS TO CC

In its most simple form, CC asks for a partition of the vertices (a *clustering*) of an unsigned and unweighted graph into clusters with the smallest number of disagreements, also called mistakes (Bansal et al., 2002). Every inter-cluster edge and every intra-cluster non-edge counts as a disagreement. This problem, also known as Cluster Editing, can be equivalently formulated on complete signed graphs by labeling every edge as positive, while introducing a negative edge for every non-edge.

CC has an equivalent formulation in terms of forbidden subgraphs, and therefore as an edge deletion problem. Namely, remove the minimum number of edges such that the re-

maining graph has no bad *cycles* (a cycle with exactly one negative edge) (Davis, 1967; Charikar et al., 2005; Demaine et al., 2006).[1] Indeed, every bad-cycle-free graph is *clusterable* (i.e., there is some partition with zero disagreements) and such partition is obtained by the connected components of the graph induced by the positive edges.

The importance of bad triangles for CC is well-recognized, as every clustering makes at least one mistake per such triangle. This dates back to the 1960s, when Davis (1967) proved that a complete graph is clusterable iff it does not have a bad triangle. The first constant-factor approximation for CC was due to a method based on packing edge-disjoint bad triangles (Bansal et al., 2002). Later, Ailon et al. (2008) used a fractional bad triangle packing argument to prove that their famous pivot algorithm yields a 3-approximation.

**Relating the CC and BTT optima.** There is a relationship between $OPT_\Delta$, the BTT optimum on a complete graph $G$, and the CC optimum $OPT_{CC}$ of $G$. The inequality $OPT_\Delta \leq OPT_{CC}$ is immediate, since the disagreements of any feasible clustering cover all bad triangles. Veldt (2022) showed that $OPT_{CC} \leq 2OPT_\Delta$. He proved this by transforming a feasible bad triangle cover into a valid clustering using pivot on an auxiliary graph. The auxiliary graph is constructed by *flipping* the signs of all edges in the cover. Mistakes are then attributed to the pivot mistakes made on the auxiliary graph, which can be charged to the number of sign flips. By using a pivot procedure with more refined inclusion probabilities, we improve this bound to $\frac{3}{2}$.

**Theorem 1.4.** *For complete graphs, it holds that $OPT_{CC} \leq \frac{3}{2}OPT_\Delta$.*

The algorithmic question of transforming a feasible bad triangle cover $F$ into a clustering with disagreements at most $\alpha|F|$ for smallest possible $\alpha \geq 1$ is interesting. In Section 5 we prove Theorem 1.4 by giving such a poly-time transformation with $\alpha = \frac{3}{2}$. If one could do this in poly-time for even smaller values, say $\alpha = 1 + \epsilon$, then our 2-approximations for BTT imply a $2 + 2\epsilon$ approximation for CC. This could be a potential strategy to close the gap between the upper bound given by the 2.06 approximation algorithm of (Chawla et al., 2015), and the lower bound of 2 on the integrality gap of the standard linear program with triangle inequality constraints (LP$_{CC}$) (Charikar et al., 2005; Demaine et al., 2006).[2]

An interesting open question is whether $OPT_{CC} = OPT_\Delta$, which we could not prove nor refute by finding numerical counterexamples.

---

[1]This formulation also holds for general graphs.

[2]Recent research has shown that CC admits approximations with guarantee smaller than two (Cohen-Addad et al., 2022), but closing this gap is still interesting.

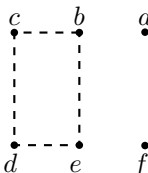

*Figure 1.* Removing edges from an optimal BTT solution can lead to a bad cycle. Dashed edges are negative edges, all other pairs are connected by a positive edge (not drawn).

**Open Question.** *For complete graphs, does it hold that $OPT_{CC} = OPT_\Delta$? If not, can one further improve the bound from Theorem 1.4?*

Figure 1 shows a graph, which after the removal of the edges from an optimal BTT solution, results in a graph that still has a bad cycle. The graph from Figure 1 satisfies $OPT_{CC} = OPT_\Delta = 4$. The edges $\{ac, ae, bf, df\}$ are a valid cover. After removing them, there are still bad cycles such as $a, b, c, f$. However, there is an alternative optimal cover, namely all the negative edges, that deletes all the bad cycles. Thus, a simple feasibility argument cannot be used to prove that $OPT_{CC} = OPT_\Delta$. Note that the question only makes sense for complete graphs. There exist non-complete graphs with many bad cycles, without having a bad triangle.

### 1.2.3. RECENT CC ALGORITHMS BASED ON BTT

Better than 3-approximations for CC were obtained by solving LP$_{CC}$ (Ailon et al., 2008; Chawla et al., 2015). One practical limitation is that LP$_{CC}$ has $\Theta(n^3)$ constraints and is thus very expensive to solve. To address this scalability issue, several authors (Veldt, 2022; Bengali & Veldt, 2023; Makarychev & Chakrabarty, 2023; Cao et al., 2024b; Balmaseda et al., 2024; Fischer et al., 2025; Veldt, 2026) have investigated how LP$_\Delta$ can be used to generate more scalable approximations for CC. This approach works well because (a) LP$_\Delta$ is a relaxation of LP$_{CC}$, and (b) approximate solutions to LP$_\Delta$ can be found quickly and combinatorially by specialized solvers for covering LPs (Cao et al., 2024b; Fischer et al., 2025; Veldt, 2026). Specifically, LP$_\Delta$ can be $(1 + \epsilon)$-approximated in time bottle-necked by finding a maximal set of edge-disjoint bad triangles (Cao et al., 2024b, Corollary 1.3). By rounding an approximate LP$_\Delta$ solution into a feasible clustering, Cao et al. (2024b) showed a poly-logarithmic depth parallel algorithm for CC with approximation ratio $2.4 + \epsilon$.

Even more simple and scalable CC approximations (at the expense of worse approximation guarantees) are obtained by directly finding feasible bad triangle covers, without solving LP$_\Delta$ (Veldt, 2022; Bengali & Veldt, 2023; Balmaseda et al., 2024). The typical way to obtain such covers is to simply pick all the edges from a maximal set of edge-disjoint bad triangles, as mentioned before. These are then subsequently

transformed into valid clusters, for example by running pivot on some auxiliary graph. This is the MatchFlipPivot routine devised by Veldt (2022), and yields a 6-approximation for CC. Using Theorem 1.1 and Theorem 1.4, we improve this to a $(3 + \epsilon)$-approximation for CC with similar runtime as MatchFlipPivot.

## 2. Other Related Work

Kortsarz et al. (2010) showed that the problem of covering triangles in unsigned graphs can be used to approximate vertex cover on triangle-free graphs, which is as hard to approximate as vertex cover on general graphs. They also provided $(k - 1)$-approximations for covering cycles of length $k$, thereby extending the results by Krivelevich (1995) from $k = 3$ to any $k > 3$.

Chalermsook et al. (2020) proved a strengthened extension of Krivelevich (1995) to multiset transversals, where a triangle should intersect at least a fixed number of edges from the multiset.

We also mention the work of Chapuy et al. (2014), which shows a method returning a triangle transversal of size at most $2x - \sqrt{x/6} + 1$, where $x$ is the optimum of the standard LP relaxation of the unsigned triangle transversal problem. This gives a $(2 - o(1))$-approximation for covering triangles in unsigned graphs. They provide similar results for unsigned multigraphs.

CC on complete graphs has been subject to a great number of algorithmic improvements recently. In their breakthrough paper, Cohen-Addad et al. (2022) went beyond a 2-approximation. Subsequently, a series of papers improved upon this, culminating in a $1.485 + \epsilon$-approximation (Cohen-Addad et al., 2023; Cao et al., 2024a; 2025) . For general graphs, CC is equivalent to Minimum Multicut and admits an $\mathcal{O}(\log n)$ approximation (Charikar et al., 2005; Demaine et al., 2006).

Adriaens & Apers (2023) studied graph property testing in signed graphs, resulting in algorithms for testing both bad-triangle-freeness and clusterability.

In Section 1.2.2 we have essentially conjectured that the optima of MinSTC+ and CC are equal on complete graphs, as we could not find a counterexample. Just like there is a connection between MinSTC+ and CC, there is also a connection between MinSTC and CD, see for example (Grüttemeier & Morawietz, 2020; Veldt, 2022). However, for these problems there are known examples where the MinSTC and CD optima differ by a factor of $8/7$ (Grüttemeier & Morawietz, 2020).

For CD, there exists an elegant LP-based 2-approximation (Veldt et al., 2018). Balmaseda et al. (2024) gave faster combinatorial approximations which scale to large graphs.

## 3. Approximation Algorithms

Here we present our approximation algorithms for BTT. All algorithms work for general graphs. The proofs of their approximation guarantees are given in Appendix.

### 3.1. Preliminaries

For a signed graph $G = (V, E^+, E^-)$ let $E = E^+ \cup E^-$. Let $T$ be the set of bad triangles. If an edge $e \in E$ is part of a bad triangle $t \in T$, then we write $e \in t$. The natural LP relaxation of BTT is given by

$$
\begin{aligned}
\min \quad & \sum_{e \in E} x_e, & \text{(LP}_\Delta) \\
\text{s.t.} \quad & \sum_{e \in t} x_e \geq 1, & \forall t \in T, \\
& x_e \geq 0, & \forall e \in E.
\end{aligned}
$$

The dual of LP$_\Delta$ is the following packing problem

$$
\begin{aligned}
\max \quad & \sum_{t \in T} y_t & \text{(PackingLP)} \\
\text{s.t.} \quad & \sum_{t : e \in t} y_t \leq 1, & \forall e \in E, \\
& y_t \geq 0, & \forall t \in T.
\end{aligned}
$$

### 3.2. BTT algorithms based on LP$_\Delta$

Algorithm 1 is the folklore 2-approximation discussed in Section 1.1, but rephrased to the BTT setting.

Algorithm 2 is based on complementary slackness properties, and is a special case of Algorithm 3 for $r = 1$.

Algorithm 3 is inspired by a randomized algorithm proposed by Lovász (1975) for finding a minimum vertex cover in $k$-partite $k$-uniform hypergraphs. To see the connection, construct a 3-uniform *bipartite* hypergraph $\mathcal{H}$ as follows. Every node in $\mathcal{H}$ corresponds to an edge in $G$. There is an edge $\{e_1, e_2, e_3\}$ in $\mathcal{H}$ if $e_1, e_2$ and $e_3$ form a bad triangle in $G$. $\mathcal{H}$ is bipartite, in the sense that every edge in $\mathcal{H}$ contains exactly one node corresponding to a negative edge from $G$. BTT is equivalent to finding a minimum vertex cover in $\mathcal{H}$. Since $\mathcal{H}$ is not 3-partite we cannot directly use Lovász' algorithm. Yet, with some modifications to the algorithm, we get a 2-approximation for BTT.

**Lemma 3.1.** *Algorithm 1 is a 2-approximation for BTT.*

**Lemma 3.2.** *Algorithm 2 is a 2-approximation for BTT.*

**Lemma 3.3.** *Algorithm 3 is a 2-approximation in expectation for BTT.*

Let us briefly compare the worst-case time complexity of Algorithm 1 compared to Algorithms 2 and 3, without going

**Algorithm 1** BTT 2-approximation by Krivelevich (1995)

> **Input:** Signed graph $G = (V, E)$ and optimal $\{x_e\}$ values from $LP_\Delta$.
> $C \leftarrow \emptyset$.
> **while** $x_e = 0$ for some $e \in E$ **do**
>     $C \leftarrow C \cup \{e \in E : x_e \geq 1/2\}$.
>     Remove $e \in E$ with $x_e = 0$ or $x_e \geq 1/2$ from $G$.
>     Solve $LP_\Delta$ on new $G$ to obtain new $\{x_e\}$ values.
> **end while**
> $V_1, V_2 \leftarrow$ 2-approximate max cut, see (Vazirani, 2001).
> $C \leftarrow C \cup \{(u, v) \in E : u, v \in V_1\}$.
> $C \leftarrow C \cup \{(u, v) \in E : u, v \in V_2\}$.
> **Return:** $C$.

**Algorithm 2** A simple deterministic 2-approximation for BTT.

> 1: **Input:** Signed graph $G = (V, E)$ and optimal $\{x_e\}$ values from $LP_\Delta$.
> 2: $E_{>0}^- \leftarrow \{e \in E^- : x_e > 0\}$.
> 3: $E_{\geq 1/2}^+ \leftarrow \{e \in E^+ : x_e \geq 1/2\}$.
> 4: **Return:** $E_{>0}^- \cup E_{\geq 1/2}^+$.

in too much detail on the time required to solve LPs. Say that $LP_\Delta$ can be solved exactly in $\widetilde{\mathcal{O}}(m^\alpha)$ arithmetic operations, where $m$ is the number of edges in $G$ and $\alpha \geq 2$ is a constant related to matrix multiplication, see (Cohen et al., 2021). In the worst-case Algorithm 1 removes one edge at a time, resulting in a total time complexity bound of

$$\widetilde{\mathcal{O}}(m^\alpha + (m-1)^\alpha + \ldots + 1) = \widetilde{\mathcal{O}}(m^{\alpha+1}).$$

*Remark* 3.4. Algorithm 3 can easily be derandomized in $\mathcal{O}(|E| \log n)$ time. Separately sort both the negative and positive edges according to their $x_e$ values. Start with a solution $C$ that corresponds to $r = 1$ in Algorithm 3, containing all the non-zero negative edges and the edges from $\{e \in E^+ : x_e \geq 1/2\}$. Iterate through the sorted list of negative edges until you encounter an edge with new LP value $z$, with value different from its predecessor. Consider a candidate solution by removing from $C$ all negative edges with LP value smaller than $z$, while adding the positive edges with value $x_e \geq (1-z)/2$ to $C$. Keep iterating through both lists and repeat a similar comparison, while maintaining track of the smallest solution found so far. After the iteration terminates, we have a solution at least as good as the output from Algorithm 3. Indeed, each possible outcome for $r \in [0, 1]$ is compared with at some point.

*Remark* 3.5. Algorithm 3 has two attractive properties. First, it also 2-approximates *weighted* BTT, which asks to minimize $\sum_{e \in E} w_e x_e$. Secondly, it gives a $(2 + 2\epsilon)$-approximation if $\{x_e\}$ is a *feasible* fractional cover with total cost $\leq (1 + \epsilon)LP_\Delta$.

**Algorithm 3** A simple randomized 2-approximation for BTT.

> 1: **Input:** Signed graph $G = (V, E)$ and optimal $\{x_e\}$ values from $LP_\Delta$.
> 2: $C \leftarrow \emptyset$.
> 3: Draw $r \in [0, 1]$ uniformly at random.
> 4: For every $e \in E^+$, add $e$ to $C$ if $x_e \geq r/2$.
> 5: For every $e \in E^-$, add $e$ to $C$ if $x_e > 1 - r$.
> 6: **Return:** $C$.

# 4. Hardness of Approximation

Several negative results concerning the approximability of BTT and related problems are discussed next. All proofs are found in Appendix.

Let us start by the following observation regarding the achievable approximation ratio for algorithms based on our linear program.

**Lemma 4.1.** *The integrality gap of $LP_\Delta$ is at least two, even in complete graphs.*

Therefore, any approximation algorithm based on $LP_\Delta$ cannot achieve competitive ratio better than two, even in complete graphs. Next we show that in general graphs a ratio of two is most likely the best one. For complete graphs it leaves open the possibility of an approximation ratio smaller than two, although Lemma 4.1 implies that such method cannot be based on rounding $LP_\Delta$.

**Theorem 4.2.** *BTT is as hard to approximate as Vertex Cover (VC), meaning that an $\alpha$-approximation for BTT can be used to $\alpha$-approximate VC.*

Khot & Regev (2008) proved that VC is UGC-hard to approximate with factor better than 2. As a consequence of Theorem 4.2, BTT is also UGC-hard to approximate with factor better than 2.

## 4.1. Complete graphs

Theorem 4.2 does not apply to complete graphs, since the reduction in its proof does not result in a complete graph. Instead, we reduce from the Minimum 2CNF Deletion problem (MD). MD asks for a variable assignment that minimizes the number of unsatisfied clauses in a 2SAT formula. Let $OPT_{MD}$ denote the optimum of MD on an instance which will be clear from context.

The following problem is shown to be NP-hard by Chlebík & Chlebíková (2004).

**Theorem 4.3** ((Chlebík & Chlebíková, 2004), Section 3.1)**.** *Let $0 < \delta \leq \frac{1}{194}$ be a constant, and let $0 < \epsilon < \delta/2$ be arbitrarily small independently of $\delta$. Given a 2CNF formula on $n$ variables, with no repeated clauses, exactly 4*

*occurrences per variable, and where each variable appears exactly once as negated, it is NP-hard to distinguish between*

(i) $OPT_{MD} < (2\delta + \epsilon)n$, and

(ii) $OPT_{MD} > (3\delta - \epsilon)n$.

We describe a gap-preserving reduction from the above problem to BTT on a *complete* graph $G$. Our reduction uses "wheel-gadgets" (specifically, *hexagrams*) similar to those used in the reduction by Charikar et al. (2005) to show APX-hardness for CC. Wheel gadgets were first introduced in the well-known reduction from 3SAT to 3-dimensional matching by Papadimitriou (2003). Our reduction differs from (Charikar et al., 2005) in two major ways. First, we reduce from the MD problem, whereas they reduce from the max 2-colorable subgraph problem in 3-uniform hypergraphs. Secondly, our clause gadgets are smaller than their hyperedge gadgets, greatly simplifying the overall analysis.

### 4.1.1. REDUCTION

In the following we describe the set of positive edges of $G$. All non-described node pairs will be connected by a negative edge.

First, we describe the node gadgets. Consider a 2SAT formula from Theorem 4.3. For each variable $z$ in the formula, create a hexagram consisting of 12 nodes, as shown in Figure 2. The hexagram related to $z$ is called the *z-hexagram*. The inner part of a hexagram consists of a cycle of length 6. Each edge of this cycle is part of one outward pointing triangle, called a *tooth*. The node of a tooth that is not part of the inner 6-cycle is called a *crown*. There are 6 crown nodes, label them as $z_i$ for $1 \leq i \leq 6$. Call a crown $z_i$ even (resp. odd) if its subscript $i$ is even (resp. odd). Likewise, call a tooth even (resp. odd) if it contains an even (resp. odd) crown node. All three even teeth of a hexagram are pairwise vertex-disjoint, and so are the odd teeth.

Next, we describe the clause gadgets. Let the $\ell$-th clause $C_\ell$ in the 2SAT formula be, for example, $C_\ell = \overline{x} \vee y$. Create a clause node $c_\ell$. Because $x$ appears negated in $C_\ell$, we create a clause edge by connecting $c_\ell$ to an odd crown $x_{odd}$ in the $x$-hexagram. Create a second clause edge by connecting $c_\ell$ to an even crown $y_{even}$ in the $y$-hexagram (even, because $y$ is not negated in $C_\ell$). For our choice of $y_{even}$, we only require that $y_{even}$ is not connected to another clause node $c_k$ for $k \neq \ell$. Do this for all clauses. Theorem 4.3 guarantees that every crown is part of at most one clause edge, since every variable in the 2SAT formula appears exactly three times as a positive literal, and one time as negated.

Figure 2 illustrates the construction corresponding to $C_\ell$. The node hexagrams and the clause edges together form the set of positive edges of our complete graph $G$. All remaining node pairs are labeled as negative.

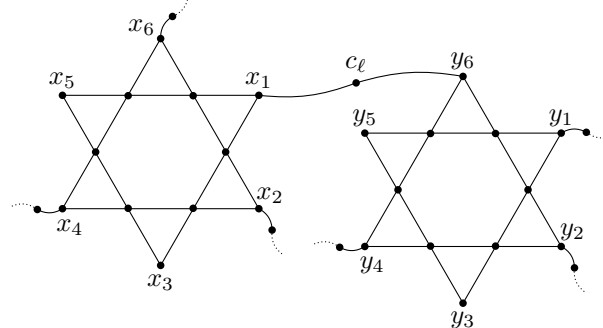

*Figure 2.* Part of the graph $G$ constructed from an instance of Theorem 4.3, if it contains a clause $C_\ell = \overline{x} \vee y$. The left (resp. right) hexagram is the node gadget corresponding to variable $x$ (resp. $y$). The $x_i$ nodes are the crown nodes of the hexagram of $x$, and similarly for $y_i$. The clause edges corresponding to clause $C_\ell$ are $(x_1, c_\ell)$ and $(c_\ell, y_6)$.

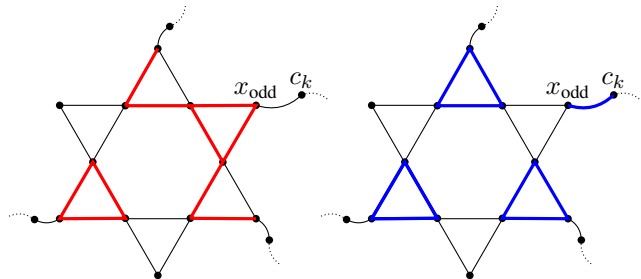

*(a)* Inconsistent cover (red).  *(b)* Consistent cover (blue).

*Figure 3.* Example to illustrate case (b) in the proof of Lemma 6.

### 4.1.2. INTUITION

The intuition is that to cover a hexagram—if one disregards bad triangles formed by clause edges for now—an optimal BTT solution will pick all 9 teeth edges from either the set of even or odd teeth. These are the only optima, all other covers consist of at least 10 edges.

The choice of odd or even teeth edges will capture whether the corresponding variable will be assigned to true or false. Clause edges will guide the variable assignment such that as many as possible clauses are satisfied. However, clause edges also form bad triangles (with hexagram edges and with each other), so we cannot simply assume that an optimal bad triangle cover only selects even or odd teeth edges.

Call a cover *consistent* if for every hexagram, the cover contains either all edges from the even teeth, or all edges from the odd teeth, and no other hexagram edges. See for example, Figure 3. We will show that there exists a consistent optimal BTT solution of $G$. This allows us to relate its optimal value to the MD optimum. By applying Theorem 4.3, we obtain a gap which leads to Theorem 4.6.

### 4.1.3. ANALYZING THE GAP

Let $OPT_\triangle(G)$ be the optimal value of BTT on $G$, and $OPT_{MD}$ be the optimal value of MD on instances from Theorem 4.3. The following two lemmas are the crux of the reduction.

**Lemma 4.4.** $OPT_\triangle(G) \leq 11n + OPT_{MD}$.

**Lemma 4.5.** $OPT_\triangle(G) \geq 11n + OPT_{MD}$.

Combining Lemma 4.4, Lemma 4.5 with Theorem 4.3 yields Theorem 4.6 after a brief calculation.

**Theorem 4.6.** *For any $\gamma > 0$, it is NP-hard to approximate BTT on complete graphs with a factor of $\frac{2137}{2136} - \gamma$.*

*Proof.* By Lemma 4.4 and Lemma 4.5 we find that $OPT_\triangle(G) = 11n + OPT_{MD}$. Using Theorem 4.3 for $\delta = \frac{1}{194}$, it follows that for any $0 < \epsilon < \delta/2$, it is NP-hard to distinguish between

$$OPT_\triangle(G) < \left(11 + \frac{2}{194} + \epsilon\right)n = \left(\frac{2136}{194} + \epsilon\right)n \quad \text{and}$$

$$OPT_\triangle(G) > \left(11 + \frac{3}{194} - \epsilon\right)n = \left(\frac{2137}{194} - \epsilon\right)n.$$

By setting $\epsilon$ sufficiently small (depending on $\gamma > 0$), the result follows. $\square$

### 4.2. Results for MinSTC, CC and CD

A very useful aspect of the previous reduction is that it also applies to three other problems: the MinSTC, CC and CD problems.

**Theorem 4.7.** *For any $\gamma > 0$, it is NP-hard to approximate MinSTC, CC and CD on complete graphs with a factor of $\frac{2137}{2136} - \gamma$.*

We briefly explain these problems and their known hardness results.

**MinSTC.** This problem is similar to BTT on complete graphs, but MinSTC requires to use only positive edges for covering bad triangles (Sintos & Tsaparas, 2014). In our reduction, we have assumed an optimal cover only uses positive edges since every negative edge is part of at most one bad triangle. This implies Theorem 4.7 for MinSTC. We are not aware of existing inapproximability results for MinSTC in the literature.

**CC.** A consistent optimal cover yields an optimal correlation clustering with the same cost, that is, $OPT_{CC}(G) = OPT_\triangle(G)$ for the constructed $G$. Repeating the argument of Theorem 4.6 leads to Theorem 4.7 for CC. APX-hardness was shown by Charikar et al. (2005), without explicitly determining a constant. The only explicit hardness ratio for CC is given by recent work of Cao et al. (2024a), who showed

NP-hardness with respect to *randomized* reductions[3] to approximate CC with ratio $24/23 - \gamma$. In contrast, our ratio is worse, but the reduction is deterministic.

**CD.** Cluster deletion (CD) is a related problem, which asks to delete the minimum number of edges of an unsigned graph such that the remaining graph is a union of vertex-disjoint cliques (Shamir et al., 2004). Note that the optimal correlation clustering for $G$ has only cliques. Consequently, $OPT_{CD}(G) = OPT_{CC}(G)$ for the constructed $G$. Repeating the argument of Theorem 4.6 leads to Theorem 4.7 for CD. Shamir et al. (2004) proved APX-hardness for CD, but without an explicitly determined hardness ratio.

## 5. From Covers to Clusters

As a final contribution, we address the following problem: Given a complete signed graph $G = (V, E)$ and a feasible bad triangle cover $F \subseteq E$, can we efficiently find a clustering with at most $\alpha|F|$ mistakes, for some small $\alpha \geq 1$?

Veldt (2022) showed that first switching (or *flipping*) the signs from the edges in $F$, and then running pivot on the resulting graph, leads to a clustering with at most $2|F|$ mistakes.

Algorithm 4 shows an alternative pivot strategy that gives at most $\frac{3}{2}|F|$ mistakes. The algorithm is similar to a standard pivot approach (Ailon et al., 2008), except when we encounter an edge in $F$. Assume $u$ is our pivot and $v$ is the node we consider adding to the corresponding cluster $P_u$. The standard pivot will add $v$ to $P_u$ if and only if $uv \in E^+$. We modify this approach if $uv \in F$. In that case, Algorithm 4 adds $v$ with probability $1/4$ if $uv \in E^+$, and with probability $3/4$ if $uv \in E^-$.

Next we state the approximation guarantee of the algorithm. Note that Theorem 5.1 implies Theorem 1.4.

**Theorem 5.1.** *Algorithm 4 returns a clustering with at most $\frac{3}{2}|F|$ mistakes in expectation.*

**Proof sketch.** Our analysis of Algorithm 4 follows the same setup as (Ailon et al., 2008; Chawla et al., 2015; Cohen-Addad et al., 2022). A key difference is that they charge the mistakes made by pivot to the LP values of the assigned edges. Instead, we assign a budget $b(u, v) = 1$ for every $uv \in F$ and $b(u, v) = 0$ if $uv \notin F$, and charge the mistakes to the budgets. Following a similar analysis, it suffices to prove that for every triplet $\{u, v, w\} \in \binom{V}{3}$ it holds that[4]

$$\frac{d(u, v|w) + d(u, w|v) + d(v, w|u)}{b(u, v|w) + b(u, w|v) + b(v, w|u)} \leq \frac{3}{2}. \quad (1)$$

---

[3] A randomized reduction technically does not show NP-hardness, but has other implications, see (Papadimitriou, 2003).

[4] In Appendix we discuss the case when the denominator is zero.

**Algorithm 4** Pivot-method for transforming a feasible BTT covers into clusters.

1: **Input:** Complete signed graph $G = (V, E^+, E^-)$ and a feasible cover $F \subseteq \binom{V}{2}$ for BTT.
2: Mark all $v \in V$ as unclustered.
3: $P \leftarrow \emptyset$.
4: **while** there are unclustered nodes **do**
5:    Pick a uniformly random pivot $u$ from all unclustered nodes.
6:    $P_u \leftarrow \{u\}$.
7:    **for** every $\{$unclustered $v \mid uv \in E^+\}$ **do**
8:       **if** $uv \in F$ **then**
9:          Add $v$ to $P_u$ with probability $1/4$.
10:      **else**
11:         Add $v$ to $P_u$ with probability $1$.
12:      **end if**
13:   **end for**
14:   **for** every $\{$unclustered $v \mid uv \in E^-\}$ **do**
15:      **if** $uv \in F$ **then**
16:         Add $v$ to $P_u$ with probability $3/4$.
17:      **else**
18:         Add $v$ to $P_u$ with probability $0$.
19:      **end if**
20:   **end for**
21:   Mark all $v \in P_u$ as clustered.
22:   $P \leftarrow P \cup \{P_u\}$.
23: **end while**
24: **Return:** Partition $P$.

Here, $d(u,v|w)$ denotes the probability that edge $uv$ becomes a disagreement when $w$ is the pivot. Likewise, $b(u,v|w)$ denotes the probability that edge $uv$ is removed from the graph when $w$ is the pivot (meaning, at least one of $u$ or $v$ is added to $P_w$) multiplied by the budget $b(u,v)$.

The proof follows from a case analysis. Because $G$ is complete, there are 4 types of triplets $\{u,v,w\}$ according to the number of positive edges in a triplet. For each type of triplet, we also have to consider all possibilities of its edges being part of $F$ or not. For each case, we verify that Eq. (1) holds. The motivation behind Eq. (1) and the full case analysis can be found in Appendix.

## 6. Conclusions

This paper makes several algorithmic contributions to the *Bad Triangle Transversal* problem in signed graphs (BTT), which asks for the minimum number of edges to be deleted from a given signed graph, so that the remaining graph does not have a triangle with exactly one negative edge. Our contributions are both positive (simpler and faster algorithms), and negative (novel hardness results).

An interesting direction for future work is to obtain a better

than 2-approximation for BTT on complete signed graphs, similarly like the recent results on Correlation Clustering (CC), or prove that such algorithm is unlikely to exist. Additionally, answering the open question posed in Section 1.2.2 would further strengthen the connection between the BTT and CC problems.

To conclude, we mention that improved gap results over those stated by Theorem 4.3 for MD on bounded-degree instances, will directly give better constants in Theorem 4.6 and 4.7. We are not aware of such results, but they might exist.

## Acknowledgements

This research is supported by the Academy of Finland project PAPADAM (371746) and by the Helsinki Institute for Information Technology (HIIT). We thank the anonymous reviewers for their helpful comments.

## Impact Statement

This paper presents work whose goal is to advance the field of Machine Learning. There are many potential societal consequences of our work, none which we feel must be specifically highlighted here.

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

# A. Proofs.

**Proof of Lemma 3.1.**

*Proof.* See (Krivelevich, 1995). The proof uses complementary slackness, the fact that one can always efficiently find a bad triangle cover that uses at most $|E|/2$ edges, and a telescoping sum argument to upper bound $|C|$ by two times the total $\text{LP}_\Delta$ cost of the initial graph $G$. $\square$

**Proof of Lemma 3.2.**

*Proof.* By the $\text{LP}_\Delta$ constraints, every bad triangle for which the negative edge $e$ has $x_e = 0$ needs to contain at least one positive edge with $x$ value at least a $1/2$. Hence, Algorithm 2 returns a feasible cover. Next, we prove the approximation guarantee. Let $\text{OPT}_\Delta$ be the optimal value. Let $T_{>0}$ be the triangles in $T$ for which the negative edge $e$ has $x_e > 0$. Let $T_0$ be the triangles in $T$ for which the negative edge $e$ has $x_e = 0$. Clearly, $T = T_0 \cup T_{>0}$. Let $\{x_e\}$ be an optimal solution to $\text{LP}_\Delta$ and $\{y_t\}$ be an optimal solution to PackingLP.

By complementary slackness we know that

$$\text{(a) If } x_e > 0, \text{ then } \sum_{t:e \in t} y_t = 1. \tag{2}$$

$$\text{(b) If } y_t > 0, \text{ then } \sum_{e \in t} x_e = 1. \tag{3}$$

Using the condition (a) we write

$$|E_{>0}^-| = \sum_{e \in E_{>0}^-} 1 = \sum_{e \in E_{>0}^-} \sum_{t:e \in t} y_t = \sum_{t \in T_{>0}} y_t. \tag{4}$$

Similarly,

$$|E_{\geq 1/2}^+| = \sum_{e \in E_{\geq 1/2}^+} 1 = \sum_{e \in E_{\geq 1/2}^+} \sum_{t:e \in t} y_t = \sum_{e \in E_{\geq 1/2}^+} \Big( \sum_{t \in T_{>0}:e \in t} y_t + \sum_{t \in T_0:e \in t} y_t \Big).$$

Condition (b) implies that if $y_t > 0$ for $t \in T_{>0}$, then $t$ contains at most one positive edge from $E_{\geq 1/2}^+$. Therefore,

$$\sum_{e \in E_{\geq 1/2}^+} \sum_{t \in T_{>0}:e \in t} y_t \leq \sum_{t \in T_{>0}} y_t. \tag{5}$$

It also holds that

$$\sum_{e \in E_{\geq 1/2}^+} \sum_{t \in T_0:e \in t} y_t \leq 2 \sum_{t \in T_0} y_t, \tag{6}$$

as each triangle $t \in T_0$ is counted at most twice. Adding Eq. (4), Eq. (5) and Eq. (6) gives

$$|E_{>0}^-| + |E_{\geq 1/2}^+| \leq 2\Big(\sum_{t \in T_{>0}} y_t + \sum_{t \in T_0} y_t\Big) = 2 \sum_{t \in T} y_t \leq 2\text{OPT}_\Delta,$$

where the last step uses LP duality. $\square$

**Proof of Lemma 3.3.**

*Proof.* Suppose a bad triangle with edges $a, b$ and $c$ is not covered by $C$. In other words, none of $a, b$ or $c$ is part of $C$. But then $x_a + x_b + x_c < 2(r/2) + 1 - r = 1$, which violates the $\text{LP}_\Delta$ constraints. Thus, Algorithm 3 returns a feasible cover. The probability that a negative edge $e \in E^-$ is part of $C$ equals

$$\text{Prob}[e \in C] = \text{Prob}[r > 1 - x_e] = 1 - (1 - x_e) = x_e$$

The probability that a positive edge $e \in E^+$ is part of $C$ equals

$$\mathrm{Prob}[e \in C] = \mathrm{Prob}[r \leq 2x_e] = \begin{cases} 1 & \text{if } x_e > 1/2, \\ 2x_e & \text{otherwise.} \end{cases}$$

By linearity of expectation, we find that the expected size of $C$ is

$$\mathbb{E}[|C|] \leq \sum_{e \in E^-} x_e + 2 \sum_{e \in E^+} x_e \leq 2\mathrm{OPT}_\Delta.$$

$\square$

**Proof of Lemma 4.1.**

*Proof.* We use the same example that is used to show an integrality gap for CC (Charikar et al., 2005). Consider a clique of $n$ nodes with only negative edges between them. Add a new vertex that is joined to every node of the clique with positive edges. An optimal fractional solution will assign a value of $1/2$ to all the positive edges. An optimal integral solution will pick $n - 1$ positive edges. The ratio $2(n-1)/n$ approaches two as $n$ increases. $\square$

**Proof of Theorem 4.2.**

*Proof.* Let $G$ be an unsigned graph for which we want to find a minimum vertex cover. Create a signed graph $H$ by labeling all edges in $G$ as negative. Add a new node $u$ to $H$ and connect it to each node in $G$ with a positive edge. We can convert a solution $F$ to BTT on $H$—which might use some negative edges—to a solution $P$ to BTT on $H$ with the following two properties: $(i)$ $P$ only contains positive edges, and $(ii)$ has size $|P| \leq |F|$. Indeed, for every negative edge $e \in F$ select either of the two positive edges that together with $e$ form a bad triangle, and add one of those positive edges to $P$. This works because every negative edge in $H$ is a part of exactly one bad triangle, whereas every positive edge in $H$ can be part of multiple bad triangles. $P$ naturally defines a vertex cover on $G$, by including in the cover the point $v$ for each edge in $(u, v) \in P$. Because $P$ is a feasible solution to BTT on $H$, this gives a valid vertex cover of $G$. Therefore, any $\alpha$-approximation for BTT leads to an $\alpha$-approximation for Vertex Cover. $\square$

**Proof of Lemma 4.4.**

*Proof.* Given an optimal assignment to MD, we construct a feasible bad triangle cover on $G$ as follows. If variable $x$ in the assignment is set to true, then include in the cover all the edges from the even teeth of the $x$-hexagram. If $x$ is false, include all the edges from the odd teeth. This requires 9 edges per variable, and covers all the bad triangles that do not contain a clause edge. Now consider a satisfied clause $C_\ell$. There are two edges incident to the clause node $c_\ell$. It suffices to include only one of these clause edges in the cover, for all the bad triangles formed by both clause edges to be covered. For an unsatisfied clause, we select both clause edges. This gives a feasible cover. Since an instance from Theorem 4.3 has $n$ variables and $2n$ clauses, we find

$$\mathrm{OPT}_\Delta(G) \leq 9n + (2n - \mathrm{OPT}_{MD}) + 2\mathrm{OPT}_{MD} = 11n + \mathrm{OPT}_{MD}.$$

$\square$

**Proof of Lemma 4.5.**

*Proof.* We show the existence of a *consistent* optimal bad triangle cover. Recall that a consistent cover is one that for every hexagram contains either all even teeth edges or all odd teeth edges (and only those). Since every negative edge in $G$ is part of at most one bad triangle, we may assume that we have an optimal cover $C_{\mathrm{OPT}}$ that only uses positive edges. We describe how to modify $C_{\mathrm{OPT}}$ into a consistent cover of equal cost.

Consider the $x$-hexagram for any variable $x$. The conditions of Theorem 4.3 state that $x$ appears exactly once as negated in the 2SAT formula. Let $C_k$ be the clause which contains the literal $\overline{x}$. Let $x_{\mathrm{odd}} \in x$-hexagram be the odd crown connected to clause node $c_k$. There are two cases, depending on whether the edge $(x_{\mathrm{odd}}, c_k)$ is part of $C_{\mathrm{OPT}}$ or not.

Case $(a)$: If $(x_\text{odd}, c_k) \in C_\text{OPT}$, then by optimality $C_\text{OPT}$ contains all even-teeth edges from the $x$-hexagram, and only those. Thus, $C_\text{OPT}$ already consistently covers the $x$-hexagram.

Case $(b)$: If $(x_\text{odd}, c_k) \notin C_\text{OPT}$, then the two $x$-hexagram edges incident to $x_\text{odd}$ must be part of $C_\text{OPT}$. Covering a hexagram with 9 edges can only be done by selecting either the even or odd teeth edges. Hence, if $C_\text{OPT}$ contains 9 $x$-hexagram edges, then necessarily $C_\text{OPT}$ must contain all odd-teeth edges from the $x$-hexagram, and only those. If $C_\text{OPT}$ contains 10 edges from the $x$-hexagram, then we can modify $C_\text{OPT}$ as follows. Delete all $x$-hexagram edges from $C_\text{OPT}$. Add both $(x_\text{odd}, c_k)$ and all even teeth edges from the $x$-hexagram to $C_\text{OPT}$. This gives a feasible cover. Since we have removed 10 edges and added 10 new edges to $C_\text{OPT}$ this remains optimal. This argument also shows that by optimality, $C_\text{OPT}$ cannot contain more than 10 edges from the $x$-hexagram. Figure 3 shows an example.

Repeating this modification for every variable $x$ will produce a consistent optimal cover. A consistent optimal cover includes 9 edges per hexagram. Consider any clause node $c_\ell$ and its two incident clause edges $(x_i, c_\ell)$ and $(c_\ell, y_j)$. When neither the tooth containing $x_i$ nor the tooth containing $y_j$ is included in the cover (in other words, when the clause is not satisfied), then a consistent optimal cover must contain *both* clause edges $(x_i, c_\ell)$ and $(c_\ell, y_j)$. Moreover, any cover must always pick one of these two clause edges, even for satisfied clauses, since $(x_i, c_\ell)$ and $(c_\ell, y_j)$ form a bad triangle with each other. There are $n$ variables and $2n$ clauses. Hence,

$$\text{OPT}_\triangle(G) \geq 9n + (\#\text{satisfied clauses}) + 2(\#\text{unsatisfied clauses}) \geq 11n + \text{OPT}_{MD}.$$

$\square$

**Proof of Theorem 5.1.** Our analysis of Algorithm 4 follows the same setup as (Ailon et al., 2008; Chawla et al., 2015; Cohen-Addad et al., 2022). A key difference with their work is that they charge the mistakes made by pivot to the LP values of the assigned edges.

Let $U_i$ be unmarked nodes at iteration $i$. Let $D_i$ be the disagreements incurred during the iteration $i$. Let $B_i$ be the number of edges in $F$ removed during the iteration $i$.

Let $X_{uvw}$ be a random boolean variable, equal to 1 iff $uv$ yields a disagreement, when $w$ is selected as a pivot. Similarly, let $Y_{uvw}$ be a random boolean variable, equal to 1 iff $uv \in F$ and is removed, when $w$ is selected as a pivot. Define

$$d(u, v \mid w) = \mathbb{E}[X_{uvw}] \quad \text{and} \quad b(u, v \mid w) = \mathbb{E}[Y_{uvw}].$$

We claim two lemmas.

**Lemma A.1.** *For any $u, v$, it holds that $d(u, v \mid v) \leq b(u, v \mid v)$.*

**Lemma A.2.** *Assume a triangle $uvw$ that is either good or contain an edge in $F$, then*

$$d(u, v \mid w) + d(u, w \mid v) + d(v, w \mid u) \leq \frac{3}{2}(b(u, v \mid w) + b(u, w \mid v) + b(v, w \mid u)).$$

Before we prove the lemmas, let us show that they prove our claim.

*Table 1.* Values for $d(u, v \mid w) + d(u, w \mid v) + d(v, w \mid u)$.

| Edge signs | which edges of the triangle are in $F$ | | | | | | | |
|---|---|---|---|---|---|---|---|---|
| | $---$ | $--+$ | $-+-$ | $+--$ | $-++$ | $++-$ | $+-+$ | $+++$ |
| $---$ | 0 | 0 | 0 | 0 | 0.5625 | 0.5625 | 0.5625 | 1.688 |
| $--+$ | 0 | 0 | 1.5 | 1.5 | 0.9375 | 1.875 | 0.9375 | 0.75 |
| $-++$ | $-$ | 1.5 | 1.5 | 1.5 | 0.5625 | 1.125 | 1.125 | 1.312 |
| $+++$ | 0 | 1.5 | 1.5 | 1.5 | 1.875 | 1.875 | 1.875 | 1.125 |

*Table 2.* Values for $b(u, v \mid w) + b(u, w \mid v) + b(v, w \mid u)$.

| Edge signs | which edges of the triangle are in $F$ | | | | | | | |
|---|---|---|---|---|---|---|---|---|
| | $---$ | $--+$ | $-+-$ | $+--$ | $-++$ | $++-$ | $+-+$ | $+++$ |
| $---$ | 0 | 0 | 0 | 0 | 1.5 | 1.5 | 1.5 | 2.812 |
| $--+$ | 0 | 0 | 1 | 1 | 1 | 2 | 1 | 2.562 |
| $-++$ | $-$ | 1 | 1 | 1 | 0.5 | 2 | 2 | 2.062 |
| $+++$ | 0 | 1 | 1 | 1 | 2 | 2 | 2 | 1.312 |

The cost of our algorithm during the $i$th iteration is

$$
\begin{aligned}
\mathbb{E}[D_i \mid U_i] &= \mathbb{E}[\mathbb{E}[D_i \mid U_i, w \text{ is a pivot}] \mid U_i] \\
&= \mathbb{E}[\mathbb{E}[\sum_{uv \in \binom{U_t}{2}} X_{uvw} \mid U_i, w \text{ is a pivot}] \mid U_i] \\
&= \frac{1}{|U_i|} \sum_{w \in U_i} \sum_{uv \in \binom{U_i}{2}} d(u, v \mid w) \\
&= \frac{1}{|U_i|} \sum_{uvw \in \binom{U_i}{3}} d(u, v \mid w) + d(u, w \mid v) + d(v, w \mid u) + \frac{1}{|U_i|} \sum_{uv \in \binom{U_i}{2}} d(u, v \mid v) + d(u, v \mid u) \\
&\leq \frac{3}{2|U_i|} \sum_{uvw \in \binom{U_i}{3}} b(u, v \mid w) + b(u, w \mid v) + b(v, w \mid u) + \frac{3}{2|U_i|} \sum_{uv \in \binom{U_i}{2}} b(u, v \mid v) + b(u, v \mid u) \\
&= \frac{3}{2|U_i|} \sum_{w \in U_i} \sum_{uv \in \binom{U_i}{2}} b(u, v \mid w) \\
&= \frac{3}{2} \mathbb{E}[\mathbb{E}[\sum_{uv \in \binom{U_t}{2}} Y_{uvw} \mid U_i, w \text{ is a pivot}] \mid U_i] \\
&= \frac{3}{2} \mathbb{E}[\mathbb{E}[B_i \mid U_i, w \text{ is a pivot}] \mid U_i] = \frac{3}{2} \mathbb{E}[B_i \mid U_i].
\end{aligned}
$$

Taking expectation over $U_i$, and summing over iterations proves the result.

We will now prove the lemmas.

*Proof of Lemma A.1.* If $uv \in F$, then $b(u, v \mid v) = 1$ and the result is trivial. Assume that $uv \notin F$. Then $uv$ will never generate disagreement, and $d(u, v \mid v) = 0$. $\square$

*Proof of Lemma A.2.* Let $p_{uw}$ be the probability that $u$ is included in $P_w$, if $w$ is a pivot.

*Table 3.* Ratios of values given in Tables 1–2.

| Edge signs | which edges of the triangle are in $F$ | | | | | | | |
|---|---|---|---|---|---|---|---|---|
| | $---$ | $--+$ | $-+-$ | $+--$ | $-++$ | $++-$ | $+-+$ | $+++$ |
| $---$ | $0/0$ | $0/0$ | $0/0$ | $0/0$ | 0.375 | 0.375 | 0.375 | 0.6 |
| $--+$ | $0/0$ | $0/0$ | 1.5 | 1.5 | 0.9375 | 0.9375 | 0.9375 | 0.2927 |
| $-++$ | $-$ | 1.5 | 1.5 | 1.5 | 1.125 | 0.5625 | 0.5625 | 0.6364 |
| $+++$ | $0/0$ | 1.5 | 1.5 | 1.5 | 0.9375 | 0.9375 | 0.9375 | 0.8571 |

Then

$$d(u, v \mid w) = \begin{cases} p_{uw} + p_{vw} - 2p_{uw}p_{vw}, & \text{if } uv \in E^+. \\ p_{uw}p_{vw}, & \text{if } uv \in E^-, \end{cases}$$

and

$$b(u, v \mid w) = \begin{cases} p_{uw} + p_{vw} - p_{uw}p_{vw}, & \text{if } uv \in F. \\ 0, & \text{if } uv \notin F. \end{cases}$$

Note that both $d$ and $b$ depends on the sign of the edges of the triangle, and whether these edges belong to $F$. The values of the triplet sums for $d$ and $b$ for all combinations are given in Tables 1–2. Note that one entry is removed, as it represents a bad triangle not covered by $F$.

The ratios

$$\frac{d(u, v \mid w) + d(u, w \mid v) + d(v, w \mid u)}{b(u, v \mid w) + b(u, w \mid v) + b(v, w \mid u)}$$

given in Table 3 prove the claim. $\qquad\square$

