# OpenReview forum: "Simple Algorithms for Bad Triangle Transversals with Applications to Correlation Clustering"
_ICML.cc/2026/Conference — ICML 2026 spotlight_

### Official Review · Reviewer_pPpA · 2026-02-23

**Soundness:** 4
**Presentation:** 4
**Significance:** 3
**Originality:** 3
**Overall Recommendation:** 5
**Confidence:** 4

**Summary:**

The authors study the bad triangle travels problem (BTT), which asks for a given signed graph (edges labeled positively or negatively) to delete a minimum sized set of edges such that the resulting graph has no bad triangle, that is, no triangle with two positive and one negative edge. The authors point out relations to other problems of interest, most notably Correlation Clustering (CC), for which is known that any (not necessarily optimal) solution F to BTT implies the existence of a Correlation Clustering of cost at most $2|F|$.
The authors strengthen this to $3/2 |F|$. Moreover, they provide two new 2-approximation algorithms to BTT with better runtime guarantees than the existing 2- and 3-approximation algorithms.
Their algorithms are complemented with lower bounds that show that
- there is no better-than-2 approximation under common assumptions
- even for the notable case of complete signed graphs we have that (a) the integrality gap is at least two and (b) BTT is NP-hard to approximate within a factor $\frac{2137}{2136} - \gamma$ for any $\gamma > 0$.

**Compliance With Llm Reviewing Policy:**

Affirmed.

**Final Justification:**

I keep my original assessment that this is an interesting, significant, sound and well-written contribution, so I support acceptance at ICML.

**Key Questions For Authors:**

none

**Limitations:**

yes

**Strengths And Weaknesses:**

The paper is very well structured and provides a clear overview of its results. The language and style of proving avoid ambiguities and make the proofs clear and easy to follow. I verified almost all proofs and found only a very few and minor issues, see "Minor Details".
While none of the proofs has very high technical depth (the most technical one seems to be the one of Theorem 8, which builds on existing techniques but applies some modifications to capture the problem at hand), they are non-trivial, provide insights into the problem and together provide a comprehensive picture.
These insights are novel and interesting and the authors do a very good job at pointing out to which other problems their findings relate and how they do. In particular, Theorem 7 generalizes an established lower bound to three other problems which are independently studied in the literature.
 Similarly, the authors place all their findings in context with the related work and state how their results compare to existing ones and thereby showcase the contributions of the paper.

Minor Details:
- Theorem 1: are the $\tilde{O}(n^2)$ steps a multiplicative factor? Maybe clarify this

- ln 310: As every variable occurs only once negated, it seems there is no need for a mechanism to ensure that no two clauses connect to the same odd crown.

- In the context of transforming feasible bad triangle covers to solutions to Correlation Clustering on complete graphs, the paper inconsistently switches between variables F and X for the cover (see Abstract, Introduction, Section 5). In particular, the first paragraph of Section 5 defines the cover as F and then refers to it as X.

- ln 681: it seems that (2n - #satisfied) is meant to actually be (2n - #unsatisfied)  or simply (#satisfied)

---

> ### Author Rebuttal · Authors · 2026-03-26
>
> A sincere thank you for carefully proof-reading all statements, this is greatly appreciated.
> We will fix the Minor Details you suggested, all of them appear correct observations.

---

> > ### Author Rebuttal · Reviewer_pPpA · 2026-03-31
> >
> > Everything is resolved as there weren't any major issues to begin with and the authors acknowledge the minor comments I had.
> >
> > I enjoyed reading the paper and I am happy you found my comments helpful

---

### Official Review · Reviewer_eSGC · 2026-03-05

**Soundness:** 4
**Presentation:** 3
**Significance:** 3
**Originality:** 4
**Overall Recommendation:** 5
**Confidence:** 4

**Summary:**

The paper studied the bad triangle traversal (BTT) problem and its application to correlation clustering. On a signed graph, a bad triangle is defined as a triangle with two edges with $(+)$ label and one edge with $(-)$ label. The BTT problem asks for an integral covering of the bad triangles, i.e., find a set of edges such that each bad triangle contains at least one edge in the returned set. The optimization goal here is to find or approximate the set of edges with the smallest size.

One approach to solve the problem is to relax the integral solution to linear programs to allow fractional covering of bad triangles, then use rounding algorithms to obtain the set of edges. The paper observed some work in the 1990s that already gave a rounding algorithm with a $2$-approximation, albeit the rounding is slow. On labeled complete graphs, the paper devised a novel rounding algorithm that gives a (randomized) $2.01$-approximation of the BTT problem in $O(m^{3/2})$ time.

There are known connections between the BTT problem and correlation clustering. Let the mistakes made by BTT be $OPT_{BTT}$, the paper proved that $OPT_{BTT} \leq OPT_{CC} \leq \frac{3}{2}\cdot OPT_{BTT}$ using a polynomial time rounding algorithm (a variant of the pivot algorithm). This would lead to a $3$-approximation for the correlation clustering problem – although not impressive since the approximation guarantees have improved way beyond $3$, this might be a good strategy to explore the rounding algorithms vs. the integrality gap in the classical clustering LP.

Finally, the paper proved several impossibility results, showing that $i).$ there is an integrality gap of $2$ for the BTT LP, which means the rounding algorithms in the paper is optimal; $ii).$ for labeled complete graphs, the apprixmation barrier is $2137/2136$ unless P=NP; $iii).$ for general graphs, the approximation barrier is $2$ unless the unique game conjecture is false.

**Compliance With Llm Reviewing Policy:**

Affirmed.

**Final Justification:**

I'm supportive and will be happy to see the paper get accepted.

**Key Questions For Authors:**

Most of my questions are embedded in the weakness section. I use this section for MISC comments:
- I think the technical language in this paper is very tailored to the TCS and math audience instead of general ML or learning theory audiences. For instance, the paper used the term “cover” in the sense of the LP cover, which means each bad triangle is covered by at least one edge (or a fractional cover such that the total “covering mass” sums up to at least $1$). However, it is very likely that such terms cannot be clear at first glance for an ML reader. Also, many lemmas are written in very succinct ways that assume a great deal of background from the reader (e.g., Theorem 4 says “BTT is as hard to approximate as vertex cover” – so what? As in, I understand what happened, but I don’t think we should assume everyone knows what this means for an ICML paper). Note that I’m actually from a very theoretical background, but accessibility is still something I deem to be very important.
- The 1.437 approximation in the STOC 2025 paper, unfortunately, had a small gap. The authors have fixed the gap and proved the new approximation ratio as 1.485. Please check and change on lines 201-202.
- Algorithms 1-3 use “Ensure” as the output of the algorithm. I’m not sure whether this is a good way to denote an output of the algorithm.

**Limitations:**

NA, no potential foreseeable negative societal impact.

**Strengths And Weaknesses:**

My general assessment of the paper is quite positive. In particular, I believe the paper has the following major **strength**:
- Simple algorithms and time efficiency improvement. The paper is able to fit all the main algorithms in the first 8 pages, and the algorithms are sufficiently simple and intuitive. The results give meaningful improvements in the time complexity for the BTT LP rounding.
- Technical soundness and depth: I briefly spot-checked the intuitions in the proofs, and it seems things are correct. I also like the reduction from 2CNF and the modified pivot algorithm to establish the relationship between $OPT_{BTT}$ and $OPT_{CC}$.
- Relevance in the literature: the paper is closely connected with a recent line of work in the literature to deepen the understanding between the bad triangles and the correlation clustering cost. Furthermore, as I mentioned in the summary, the paper gives a good potential strategy to close the integrality gap in the classical clustering LP.

Some **weaknesses** for the authors to consider and revise:
- The main algorithm contribution, which is a 2-approximation of the solution of the BTT problem, has no improvement over the cited algorithm in the approximation factor. This is somewhat alleviated by the simplicity of the algorithm and better rounding (post-processing) time. However, these properties were not mentioned in the lemma statements in section 3 (what are the respective time complexity bounds?).
- Similarly, for the correlation clustering lower bound, the best current lower bound already gave $1.04$-approximation for NP-hardness, which beats the lower bound proved in Theorem 7.
- Relevance to only an older version of clustering LP. The authors also noted that recent advances in correlation clustering have deviated from the LP based on bad triangle packing. Therefore, the scope of the contribution is inherently limited (although still meaningful).
- Finally, the paper does not contain experiments. This is OK from my perspective, but I want to flag this for the AC in case the conference explicitly demands experiments. I personally am not letting this factor influence my judgment.

---

> ### Author Rebuttal · Authors · 2026-03-26
>
> A sincere thanks for carefully reading our paper, and providing constructive, appreciative and knowledgeable feedback. Here are some answers to your questions:
>
> > "...  However, these properties were not mentioned in the lemma statements in section 3 (what are the respective time complexity bounds?)"
>
> Good point, we will add them.
> The triangle cover LP has $m = |E|$ variables, one for each edge. The number of constraints are the number of bad triangles.
> Say the triangle covering LP can be solved in $O(m^{\alpha})$ time for some $\alpha \geq 1$ ($\alpha=2.38$ should be safe, although the exponent $\alpha$ for *covering* LPs might be smaller, need to carefully check the literature). Then in the worst-case Algo1 iteratively removes one edge at a time, and Algo1 needs to solve a cover LP for $O(m)$ graphs, resulting in something like $O(m^{\alpha}+(m-1)^{\alpha}+\ldots+1)$. This would be make the total running time $O(m^{\alpha+1})$. This is a very rough analysis though.
>
> > "Similarly, for the correlation clustering lower bound, the best current lower bound already gave 1.04-approximation for NP-hardness, which beats the lower bound proved in Theorem 7"
>
> Agreed, but they show the NP-hardness only under randomized reductions. Our reduction is deterministic.
>
> > "Relevance to only an older version of clustering LP. The authors also noted that recent advances in correlation clustering have deviated from the LP based on bad triangle packing. Therefore, the scope of the contribution is inherently limited (although still meaningful)."
>
> We fully agree with this comment.
>
> > "Finally, the paper does not contain experiments"
>
> See our answer to Reviewer L6c1.
> We will also fix the MISC comments, thanks for pointing these out.

---

> > ### Author Rebuttal · Reviewer_eSGC · 2026-03-31
> >
> > Thank you for getting back to me about the comments. I hope my comment can help the authors strengthen the paper. As I said, I'm supportive and will be happy to see the paper get accepted.

---

### Official Review · Reviewer_L6c1 · 2026-03-13

**Soundness:** 3
**Presentation:** 3
**Significance:** 3
**Originality:** 3
**Overall Recommendation:** 5
**Confidence:** 3

**Summary:**

The work considers the problem of identifying a minimum traversal for bad triangles in signed graphs (BTT for short). Based on LP techniques, two new 2-approximation algorithms are designed, one of which is randomized. Several hardness results are discussed, including the integrality gap and the hardness of approximation for complete graphs. Several connections for BTT and similar/related problems are explored, including correlation clustering and strong triadic closure. Finally, the proposed techniques are used to cluster signed graphs under the method of Veldt et al. (2018).

**Compliance With Llm Reviewing Policy:**

Affirmed.

**Final Justification:**

The authors' rebuttal addressed most of my concerns. I am positive towards the work.

**Key Questions For Authors:**

Consider weaknesses and minors

**Limitations:**

Yes

**Strengths And Weaknesses:**

Strengths.
S1. The considered algorithmic problem is challenging and important.
S2. The techniques are simple and well-explained.
S3. There are several interesting connections with many problems.


Weaknesses.
W1 Applicability outside complete graphs.
Several results of the work only apply to complete graphs (lower bound and some relations to other algorithmic problems). I find this to limit the applicability of the contribution in practical settings.

W2 Deterministic vs randomized algorithm.
Algorithm 3 is randomized; this is not clear in the contribution list. The authors discuss derandomization, which incurs an extra cost compared to existing methods. It would be nice to have a clear and detailed Table showing the complexity/approximation ratios of existing methods and proposed ones.

W3. There are no experiments.
It would be nice to validate the claimed “simpler and faster” approaches in practice. Especially given the wide literature/algorithms available for the considered problem.

Minors.
- BTT has different fonts in the abstract.
- Line 55: to be the correct -> to be a tight lower bound.

---

> ### Author Rebuttal · Authors · 2026-03-26
>
> A sincere thank you for your feedback, and general appreciation of the paper. Here are our answers to your weaknesses and concerns:
>
> > "W1 Applicability outside complete graphs.
> Several results of the work only apply to complete graphs (lower bound and some relations to other algorithmic problems). I find this to limit the applicability of the contribution in practical settings."
>
> A: You are correct in saying that, for general signed graphs, there is no strong connection between BTT and the two other problems discussed in the paper (CC and STC+).
> Also, the algorithmic positive/negative results in this setting are essentially settled: Theorem 4 states that a better than 2-approximation is unlikely, and Algo 1/2/3 give such a 2-approximation, so there is not much room for improvements.
>
> The problem becomes more interesting on complete signed graphs for two reasons: a) there is a strong connection between BTT and with STC+ and CC, and b) it leaves room for better algorithmic results.
> We do not necessarily agree that our BTT results for complete signed graphs limits its applicability in practice. Complete signed graphs are essentially just normal (unsigned) graphs.
> CC on complete signed graphs (and also to a lesser extent STC+) is an extremely well-studied and fundamental clustering problem in ML.
> Therefore, drawing connections between the two problems is meaningful, also in practice.
> We tried to generalize our results as much as we can (if it also holds for general signed graphs, we state it), but if it only holds for complete signed graphs, then this is not necessarily a big disadvantage in our opinion.
>
>
>
> > "W2 Deterministic vs randomized algorithm. Algorithm 3 is randomized; this is not clear in the contribution list. The authors discuss derandomization, which incurs an extra cost compared to existing methods. It would be nice to have a clear and detailed Table showing the complexity/approximation ratios of existing methods and proposed ones."
>
> Thanks for pointing this out. We will fix this in Section 1.3, and add a Table with runtimes/approx. for each method (we have space for this).
>
> > "W3. There are no experiments. It would be nice to validate the claimed “simpler and faster” approaches in practice. Especially given the wide literature/algorithms available for the considered problem."
>
> We did some preliminary experiments before submission, using a standard commercial LP solver (Gurobi in python) to solve the triangle covering LP.
> The findings are coherent; our Algo2 and Algo3 were faster than the baseline Algo 1, while also resulting in better (i.e., smaller-sized) solutions.
> They were slower (not-surprisingly) than the standard 3-approximation (find a maximal set of edge-disjoint bad triangles and include all the edges in the cover), but give significantly smaller-sized covers.
> The issue was that off-the-shelf solvers only managed to run on really small graphs (graphs with a few hundred nodes), and often ran out of memory, in particular for complete signed graphs.
> We did not include our experimental findings because a) the graphs were so small-sized, and b) we believe the theoretical contributions are significant and interesting enough as is.
>
> In the meanwhile, we have managed to scale things up, especially for $(1+\epsilon)$-approximating the LP.
> If accepted, we will send this work for a journal extension that contains a comprehensive experimental evaluation of our methods.
>
>
>
> > "Minors..."
>
> Fixed. Thank you for finding these.

---

> > ### Author Rebuttal · Reviewer_L6c1 · 2026-04-03
> >
> > Thank you for your rebuttal. I remain positive towards the submission, as it provides a nice contribution for the related field.

---

### Decision · Program_Chairs · 2026-04-30

**Decision:**

Accept (spotlight)

**Comment:**

This paper is about the bad triangle traversal problem, which is closely related to correlation clustering. Its focus are approximation algorithms for the problem.

The reviewers all liked the paper and agreed that it is a good contribution to the field. They appreciated the importance of the problem, as well as the simplicity of the techniques and the good writing. Insightful connections are made between various problems. The paper augments its positive results with meaningful negative ones (integrality gap, APX-hardness, unique games hardness).